# Adaptive Extremum Seeking Control of Urban Area Wind Turbines

Felix Dietrich [1,†], Steffen Borchers-Tigasson [1,†], Till Naumann [2] and Horst Schulte [1,*,†]

1   Automation and Intelligent Systems Group/Control Engineering Group, Faculty 1: School of Engineering—Energy and Information, University of Applied Sciences (HTW) Berlin, 12459 Berlin, Germany; felix.nathan.dietrich@posteo.net (F.D.); steffen.borchers@htw-berlin.de (S.B.-T.)
2   MOWEA—Modulare Windenergieanlagen GmbH, Storkower Str. 115A, 10407 Berlin, Germany; t.naumann@mowea.world
*   Correspondence: schulte@htw-berlin.de; Tel.: +49-30-5019-3301
†   Current address: Wilhelminenhofstraße 75A, 12459 Berlin, Germany.

**Abstract:** Maximum-power point tracking of wind turbines is a challenging issue considering fast changing wind conditions of urban areas. For this purpose, an adaptive control approach that is fast and robust is required. Conventional approaches based on simple step perturbations and subsequent observation, however, are difficult to design and too slow for the demanding wind conditions of urban areas including gusts and turbulence. In this paper, an extremum seeking control scheme to the recently developed wind turbine MOWEA (Modulare Windenergieanlagen GmbH) is proposed and successfully applied. To this end, a comprehensive aero-electromechanical model of the wind turbine under study including basic control is formulated. Next, the extremum seeking control scheme is adapted to the system. Several aspects to increase adaptation speed are highlighted, including a novel phase compensation. Finally, a validation of the proposed approach is performed considering real wind data, thus demonstrating its fast and robust adaptability. The proposed control scheme is computationally efficient and can be easily implemented on the existing onboard electronics.

**Keywords:** adaptive control; control of renewable energy resources; extremum seeking control

## 1. Introduction

With a rising number of countries committing to ambitious zero carbon emission goals within the next decades, increasing the share and output of renewable energy sources is key to attaining these goals.

Small wind turbines complement large-scale wind turbines as being flexible, economic, and relocalizable power modules. They are beneficially deployed where wind is available though space is limited. For wind turbines in general, the engineering challenge is to harvest the maximum power given a wide range of uncertain environmental parameters and disturbances [1], particularly continuously changing wind speed, wind direction, and local turbulence streams.

The power extracted by a wind turbine not only depends on these environmental factors but also is influenced by certain control actions. Particularly, the turbine speed as a function of generator torque constitutes an important controllable input. Environment and control together define the specific power map [2], which has a unique maximum power point with respect to turbine speed at each level of wind speed.

The goal of harvesting maximum power is referred to as maximum power point tracking. The key approach here is to optimize the power output given the current environmental conditions using optimization or, more generally, adaptive control approaches. Conventional adaptive control approaches perform a perturbation and observe techniques by repeated simple step perturbations and by subsequently monitoring the gradient of the power output, see e.g., [3,4]. Most of these techniques derived are based on discrete

analysis and require carefully balancing amplitude, direction, and frequency of the perturbation with respect to the changes in environmental parameters and the response time of the system [5]. Alternatively, wind speed can be observed using dedicated sensors, such as LIDAR (Light Detection and Ranging) systems [6] or introducing model-based observers, see e.g., [7] to control the system using a static look-up power map. This approach is either expensive or computationally demanding and, hence, is not applicable to small-scale wind turbines.

Extremum seeking provides a promising alternative approach for solving the maximum power point tracking problem in small wind turbines. It is a model-free, real-time optimization approach and is particularly well suited for the case when the maximum power point shifts continuously. Extremum seeking employs a continuous harmonic perturbation and uses the measured response to estimate the sign and amplitude of a gradient of the power map to adapt the control input optimally. Extremum seeking is well suited for dynamical systems, providing that harmonic perturbation does not excite the transient behavior of the system. This can be easily achieved by time-scale separation taking the time constants of the dynamic subsystem into account and choosing the harmonics frequency accordingly. Extremum seeking also benefits from rigorous convergence results and is thus provably stability, as shown in [8].

Extremum Seeking Control (ESC) is a dynamic research field in both method development and adoption to specific wind energy conversion systems. In [9], a logarithmic power feedback was proposed as input for the extremum seeking algorithm to cover a wider range of wind velocities. The approach is tested and validated on a large wind turbine in [10], where the average settling time is approximately 30 min. In [11], a sliding mode extremum seeking method was proposed using invasive weed optimization for determining optimal control parameters. The approach reduces steady-state oscillations; however, the approach requires fine-tuning. In [12], the sliding mode extremum seeking approach was generalized to nonlinear systems. In [13], issues of undesirable convergence under fluctuating wind of extremum seeking were addressed by using an estimated power coefficient obtained by an additional wind speed nacelle anemometer measurement in combination with extremum seeking. In [14], a model-based recursive Gaussian process approach was developed for a lighter-than-air wind energy conversion systems and was compared with extremum seeking.

More specific applications were considered in [15]; optimal adaptation for general wind energy conversion systems was achieved via estimating the optimum value of the $c_P$ coefficient using extremum seeking. In [16], the ESC was equipped in a prototype, the Delft Offshore Turbine (DOT) with a retrofitted 500 kW hydraulic drive train. In [17], a wind turbine composed of a two-mass drive train was considered. Here, indirect field orientation control in combination with ESC was applied.

Recently, the extremum seeking scheme has also been applied to arrays of wind turbines in [18,19]. By comparing individual and nested ESCs, where the latter coordinates single controllers to seek a farm-level optimum, individual control was shown to be more appropriate for sites with wind conditions changing on a short time scale while nested control is preferred when the wind conditions are quite stable. In both cases, extremum seeking increases the power production of the array.

In this paper, the extremum seeking algorithm is demonstrated to be very well suited for small wind turbines in urban areas. This is the first real-world adaptation of ESCs to a small wind energy conversion system and tested on real wind data. We particularly focus on balancing fast and robust adaption. To increase the adaptation speed while maintaining robustness, we introduce a novel phase shift compensation for the perturbation-induced system response.

In this contribution, in Section 2, the technical scheme of the small wind turbine, its mathematical model, and internal control design are formulated. In Section 3, the tailored extremum seeking algorithm, which compensates for undesired phase shift, is proposed,

and Section 4 provides the results given ideal and real wind data. Section 5 closes the paper with discussion and an outlook.

## 2. Control Objectives and Process Description

### 2.1. Control Objectives

The operation of wind turbines is broadly divided into two regions. Below the so-called design wind speed $v_d$, the rotor speed $\omega_r$ is less than the rated rotor speed $\omega_{r,R}$. This region is called the partial load region because the turbine power is less than the rated power $P_{r,R}$. Therefore, in the partial load region, the main control objective is power optimization. If the wind speed exceeds the design wind speed, the power is actively limited to $P_{r,R}$ of the rotor. Therefore, this region is called the full load region. How this is done for real wind turbines will be explained in the following. Normally, for the wind turbine power class considered in this paper ($P_{r,R} < 2$ kW), the objectives power optimization/power limitation are achieved by just controlling the generator torque $T_g$.

### 2.2. Aerodynamics of the Wind Turbine Rotor

The power characteristics of an uncontrolled wind turbine results from the aerodynamics of the rotor in free air flow. According to the stream tube theory, the wind power $P_w$ of a free air flow along the stream tube of radius $R$ with wind speed $v$ is given by

$$P_w = \frac{1}{2} \rho \, v^3 \, \pi \, R^2 \,, \tag{1}$$

where $\rho$ denotes the air density and $R$ denotes the rotor radius. Placing the rotor inside the stream tube, the power $P_r$ generated with the rotor from the wind power $P_w$ depends on the aerodynamic efficiency of the rotor, the so-called power coefficient $c_P$:

$$P_r = P_w \, c_P(\lambda) = \frac{1}{2} \rho \, v^3 \, \pi \, R^2 \, c_P(\lambda) \,, \tag{2}$$

where $v$ is the wind speed far in front of the rotor and $\lambda$ is denoted as the tip speed ratio, which is defined by

$$\lambda = \frac{\omega_r \, R}{v} \,. \tag{3}$$

From rotor power $P_r = T_r \, \omega_r$, the rotor torque $T_r$ can also be deduced:

$$T_r = \frac{1}{2} \rho \, v^2 \, \pi \, R^3 \, \frac{c_P(\lambda)}{\lambda} \,. \tag{4}$$

The $c_P$ coefficient curve, also called the $c_P$-$\lambda$ curve of the example wind turbine to be investigated, is shown in Figure 1. This curve illustrates a typical non-monotonic shape with a maximum in the middle and $c_P$ values decreasing to zero for small and large $\lambda$. Negative $c_P$ values result according to (3) because the power flow must be reversed for a further increase of the rotor speed and/or the reduction of wind speed. According to Betz [20], the theoretically maximum value that can be achieved is $c_{P,max} = \frac{16}{27}$. Note that this $c_{P,max}$ value is only achieved for an ideal wind turbine rotor without losses resulting from

- profile losses at the rotor blade sections caused by drag forces;
- flow around the blade tip, the so-called tip losses; and
- wake losses due to down stream wake rotation.

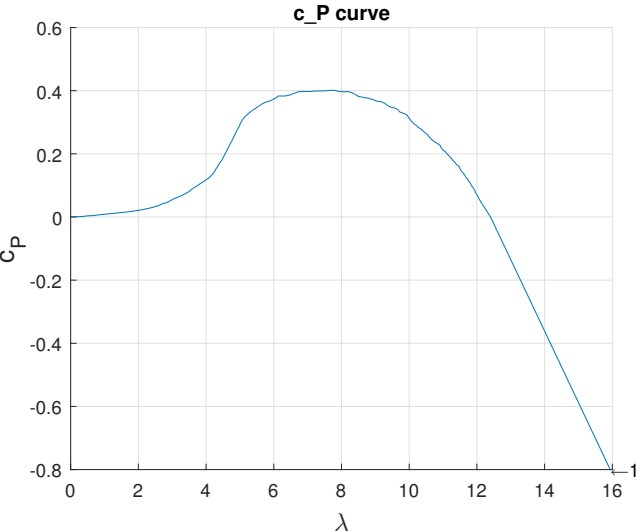

**Figure 1.** $c_P$-$\lambda$ curve of the MOWEA (Modulare Windenergieanlagen GmbH) wind turbine.

These effects are much stronger for small turbines than for large ones. For this reason, the practically achieved value of the MOWEA wind turbine according to Figure 1 is

$$c_{P,max} = c_P(\lambda_{opt}) = 0.4 \quad \text{for} \quad \lambda_{opt} = 7.5. \tag{5}$$

*2.3. Drive Train Dynamics*

To achieve optimal performance, $\omega_r$ must be set by the generator torque $T_g$ in such a way that $\lambda$ corresponds to the value at which the maximum $c_P$ value is reached:

$$\max_{\lambda(\omega_r)} c_P(\lambda) = c_P(\lambda_{opt}) = c_{P,max}, \quad \lambda_{opt} = R \frac{\omega_r}{v}. \tag{6}$$

This means that the optimum must be set dynamically by varying the rotor speed due to the generator torque without knowing the wind speed. For a more detailed study, let us consider the equation for motion of the drive train. For small wind turbines, the rotor speed is sufficiently high such that a gearbox is usually not required.

$$\dot{\omega}_r = \frac{1}{J} \left( T_r(\omega_r, v) - T_g \right), \tag{7}$$

where $v$ denotes the unknown wind speed far in front of the rotor, and $J$ denotes the total inertia of the drive train. The inertia $J$ summarizes the mass inertia of the shaft, hub, generator, and the mass inertia of the rotor blades around the rotor axis. Based on the reasonable assumption of a stiff shaft, the rotor speed is equal to the generator speed:

$$\omega_r = \omega_g, \tag{8}$$

where the considered drive train is shown in Figure 2.

*2.4. Generator Model and Current Control*

The used generator in the considered small wind turbine was based on a brushless DC (BLDC) motor. The BLDC motor operated as a generator, whereby the power flow was reversed. The alternating current caused by the induced voltage in the three stator windings was rectified by a controllable inverter. The converter also contained a boost circuit to increase the DC voltage and to continuously adjust the current in the stator on the DC side, denoted as $i_{DC}$. Due to the high dynamics of the power electronics, it is sufficient to use a simplified mean value model, where the delay due to the capacitor is also neglected

compared to the dominant inertia of the drive train. Simplified, the following generator model

$$T_g = \frac{1}{2\pi} k_M \psi i_{DC} \tag{9}$$

with an current control loop can be used:

$$\frac{di_{DC}}{dt} = -\frac{1}{\tau_i} i_{DC} + \frac{1}{\tau_i} i_{DC,ref}, \tag{10}$$

where $k_M$ denotes the motor parameter, $\psi$ denotes the constant flux caused by the permanent magnets of the rotor, and $\tau_i$ is the time constant of the current control loop based on a classic digital PI controller.

### 2.5. Cascaded Control Scheme with Extremum Seeking

The proposed cascaded control scheme for small wind turbines in the partial load with power flow is illustrated in Figure 2. Shown is the flow from the harvested wind power of the rotor $P_r$ in (2) via the generator with $P_{AC}$ and converter into the DC circuit and finally into the load denoted as $P_{DC}$.

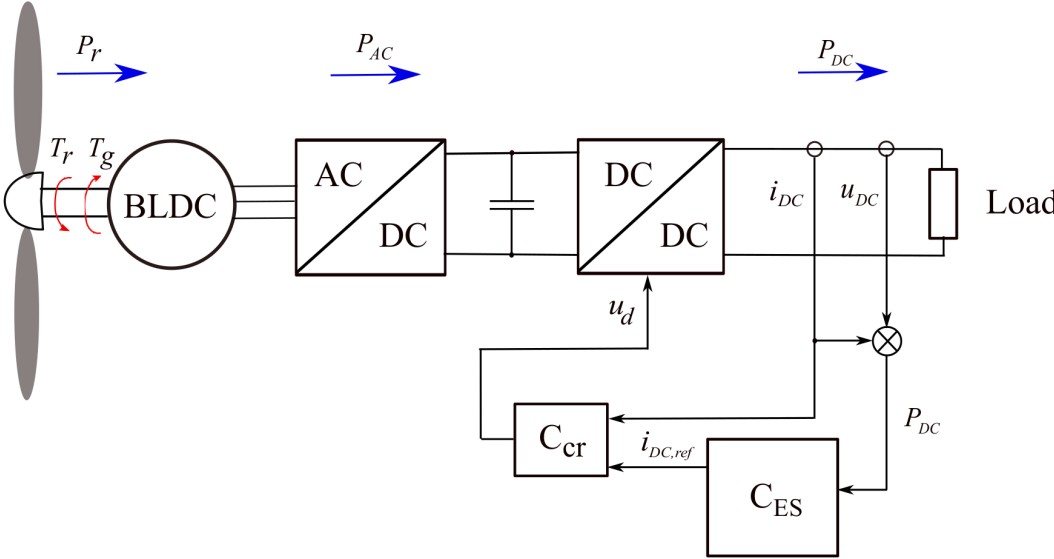

**Figure 2.** Drive train, power electronics, and control scheme with extremum seeking for small wind turbines in the partial load.

The control system structure is cascaded with an inner current controller $C_{cr}$ and the outer circuit with the Extremum Seeking (ES) method that is included in $C_{ES}$. For maximum power tracking with the ES controller, the instantaneous power with $P_{DC} = u_{DC} i_{DC}$ is measured at the load. Based on the method described in Section 3, a reference current $i_{DC,ref}$ is given for the inner current control loop. The output variable of the controller is a duty cycle signal $u_d$, which is used to set the DC/DC voltage ratio and, thus, the current ratio in reverse. The dynamics of the inner control loop can be described in a suitable form with (10). The change in $i_{DC}$ is used to adjust the generator torque according to (9), and thus, the decrease/increase in the rotor speed described by the motion Equation (7) is modified in such a way that the maximum power is tracked for variable wind speed below

the design wind speed $v_d$. The considered MOWEA wind turbine has a design wind speed of $v_d = 12\,\text{m/s}$.

## 3. Methods

### 3.1. Extremum Seeking Control

Extremum seeking is a model-free, continuous domain adaptive control approach with provable stability [8]. Examining the electrical power output map $P_{DC}$ as function of the control input $i_{DC,ref}$, a unique extremum for a given wind speed comparable to the $c_p$-$\lambda$ curve as depicted in Figure 1 is observed. The extremum is however not known a priori, and in normal operation mode, the extremum shifts permanently due to the ever-changing wind speed. Our overall objective is to seek the extremum by adapting the control input $i_{DC,ref}$ to generate maximum power $P_{DC}$.

To this end, an Extremum Seeking (ES) scheme as depicted in Figure 3 is applied. First, a persistent harmonic disturbance signal $i(t) = \hat{i}\sin\omega_p t$ is introduced and superimposes this signal onto $i_{DC,ref}$, the control input set point controlling the rotational speed. This allows us to probe the gradient of the system output $P_{DC}$. Secondly, the system response with respect to the added perturbation is extracted using a high-pass filter and the gradient is evaluated by multiplication and by considering phase compensation. Third, the gradient thus obtained is amplified and controlled to zero using the integral controller.

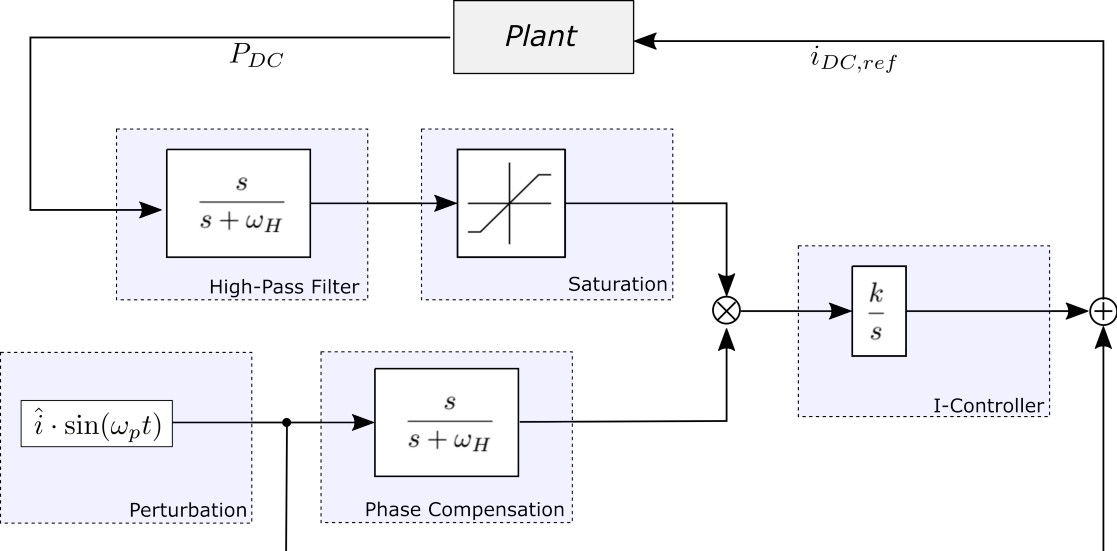

**Figure 3.** Extremum seeking control scheme. The generated power $P_{DC}$ is probed by the introduced harmonic perturbation. The response's gradient is retrieved using a high-pass filter, evaluated considering a phase shift, and controlled to zero.

### 3.2. Control Parameters

Fortunately, the proposed ES control scheme contains only a few parameters, though they have to be chosen mindfully. To ensure fast adaptation, as required for optimally harvesting energy as well as overall robustness, the dynamics of the underlying electromechanical system as well as the characteristics of the disturbances (wind changes) have to be taken into account. The maximum time constant of our electromechanical system can be estimated to approximately $\tau_i \approx 2$ s; thus, the frequency of perturbation $\omega_p = 1/4\,\text{s}^{-1}$ is chosen sufficiently slow so that no dynamic mode of the turbine is excited. The amplitude of perturbation is chosen as small as possible ($\hat{i} = 0.1$ A) considering noisy data and demanding a clear systems response in the operational space. Note that it is possible to schedule the amplitude over the operational space (e.g., log-wise as proposed in [9]) to account for nonlinear input–output characteristics though this is, for simplicity, omitted here.

Next, the gradient of the system output is extracted using a high-pass filter. Its cutoff frequency is chosen to enable passing the perturbation frequency and to cut off lower

frequencies, i.e., $\omega_H \ll \omega_p$. A saturation block is introduced to account for disruptive changes of wind conditions as observed in urban areas to a maximum level of $\pm 2$ W.

Lastly, the gain in integral controller $k$ (see Figure 3) is fitted to ensure fast and robust adaptive control for all available urban wind scenarios.

In summary, the parameters for the ESC have been chosen to trade-off fast and robust adaption. Increasing the speed of the controller further would be accompanied with decreases in overall robustness. The presented controller can, for example, handle a cold start, which may not be the case for a faster tuning set. If slower tuning is considered, the robustness margins would increase; however, overall performance would be affected.

The simulations were performed using Simulink® Mathworks R2020a.

## 4. Results

To test, analyze, and validate our extremum seeking approach, three wind disturbance regimes were considered: First, nominal wind speed regimes were tested including linear slopes and steps. Second, noise was added to the nominal regime. Finally, our approach was validated considering real wind data.

To simplify analysis of the proposed controller, "slow" wind changes were defined by passing three Beaufort scales within one minute. This is equivalent to a wind rate change of

$$\frac{dv}{dt} \approx \pm 6 \frac{m}{s} \cdot \min^{-1} = \pm 0.1 \frac{m}{s^2}.$$

In contast, "fast" wind changes are defined as wind rate changes above this threshold (absolute value).

### 4.1. Nominal Wind Regime

The nominal wind regime is visualized for a time frame of 300 s, and the results are depicted in Figure 4. The upper graph shows the wind velocity $v$ (disturbance) applied to the plant. To evaluate the overall control performance, a $c_P$ value normalized with respect to $c_{P,max}$ (5) is utilized.

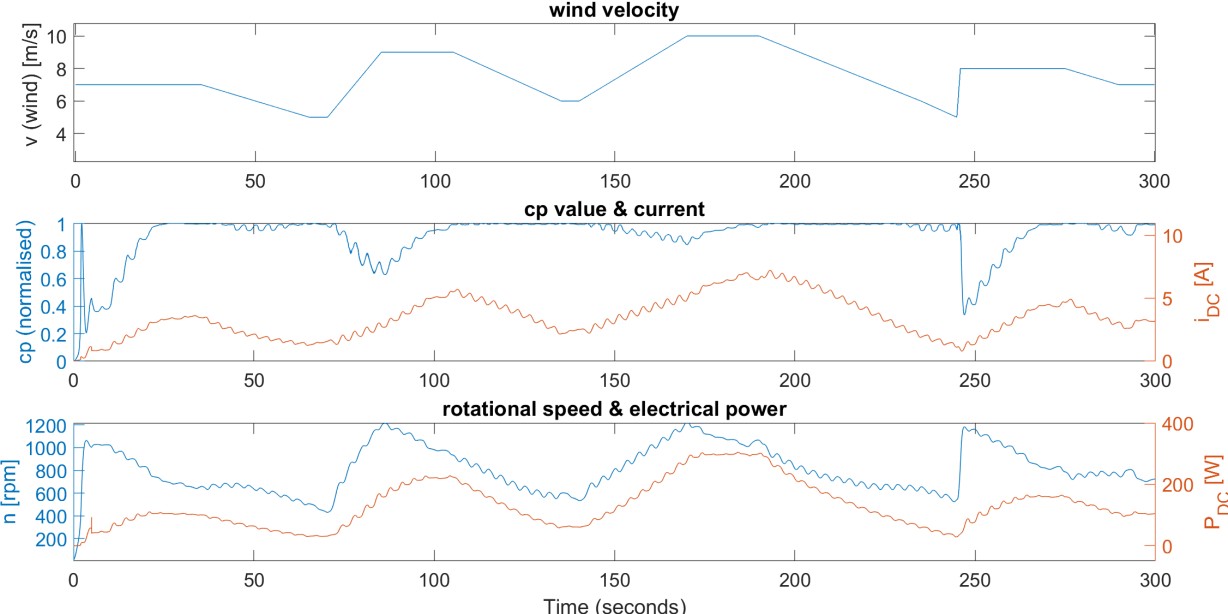

**Figure 4.** Simulation results with a nominal wind regimen composed of slow and fast changing wind velocities.

A $c_P$ value of one indicates a reached maximum power point, i.e., the controller operates optimally for the given wind velocity. A normalized $c_P$ value is shown in the

middle graph (blue). Starting from the idle state at $t = 0$ s, the perturbation signal is applied to the current $i_{DC}$, as shown in red. Note the ripples in the course of the current as a result of the introduced perturbation. It takes the system 25 s to reach maximum power point the first time. As expected, the controller operates close to the optimum for constant or slow changing wind velocities (e.g., 25 s $< t <$ 75 s). For fast changing wind velocities and disruptive gusts, the controller temporarily operates below the optimum but quickly reattains the optimal power point. In the lower graph, the corresponding power output (red) and the rotor speed (blue) are depicted.

### 4.2. Analysis Response Time

Next, the controller response is analyzed for "slow" and "fast" wind changes; see Figure 5. For slow wind changes (green), the controller adapts almost instantaneously. For faster wind rate changes (blue and red), the controller deteriorates as long as the rate change persists. Thus, for "slow" wind changes, our ES controller is fast enough.

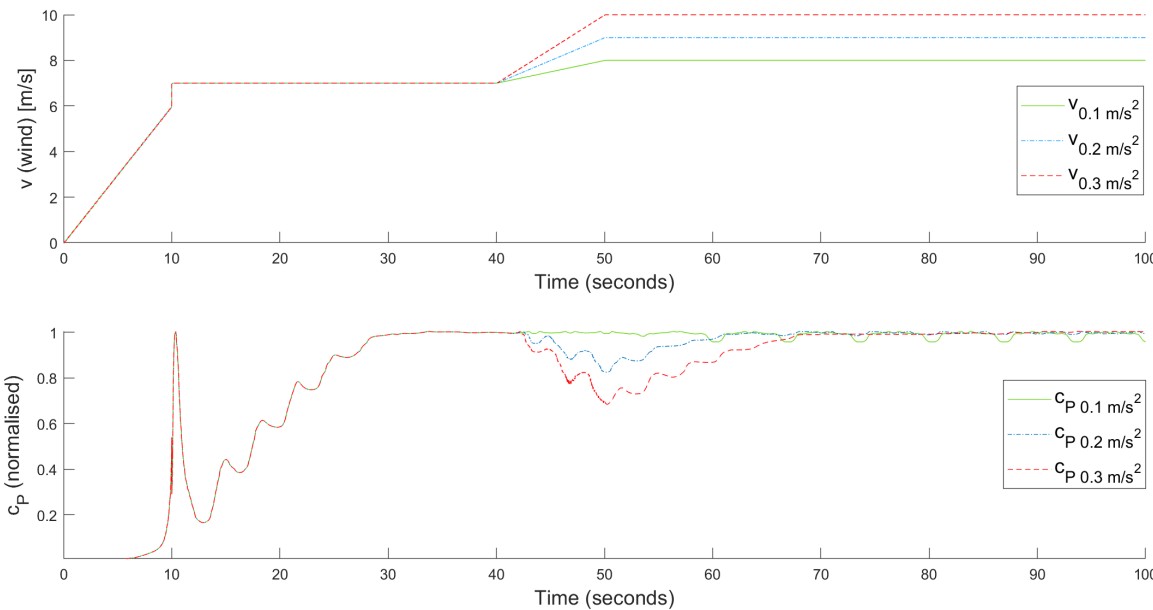

**Figure 5.** Response of the extremum seeking controller for slow (green and solid) and faster (blue or red, and dashed) wind slopes.

The response time of the ES controller was estimated using step perturbation. The relevant time constant is the time for the systems step response to reach 63.2% of its final (asymptotic) optimal value $c_P = 1$. For an average baseline wind velocity of $6 \frac{m}{s}$ and step size of $3 \frac{m}{s}$, the controller response time is estimated to $\approx$14 $s$. However, the system is nonlinear and the response time increases with the baseline as well as step size. For example, for the same baseline with an increased step size of $5 \frac{m}{s}$, the controller response time is estimated to $\approx$24 $s$.

### 4.3. Noisy Wind Regime

The noisy wind regime is displayed in Figure 6. To cover possible measurement errors, a white noise with zero mean and standard deviation is used. As shown, the extremum seeking controller performs very well considering high-frequency noise. After the study with ideal artificial data, real wind regime data can now be used.

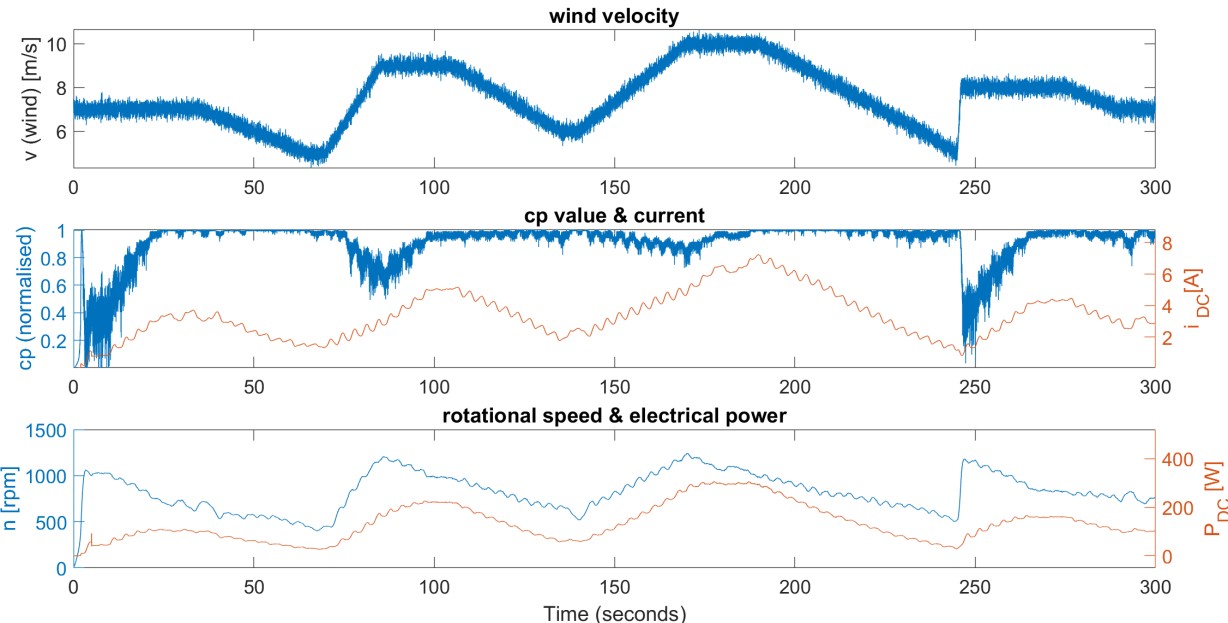

**Figure 6.** Simulation results with white noise added to the wind regimen to cover possible measurement errors.

### 4.4. Validation of Real Data

The data for validation shown in Figure 7 were collected at Kaiser-Wilhelm-Koog on the North Sea coast at an altitude of 10 m. As shown in the upper graph, the real wind data combines simultaneously gusts as well as slow and disruptive changes in wind velocity. As confirmed with a Shapiro–Wilk test (significance $\alpha = 1 \cdot 10^{-3}$), the wind rate changes are normally distributed (mean 0.003 and standard deviation 1.30).

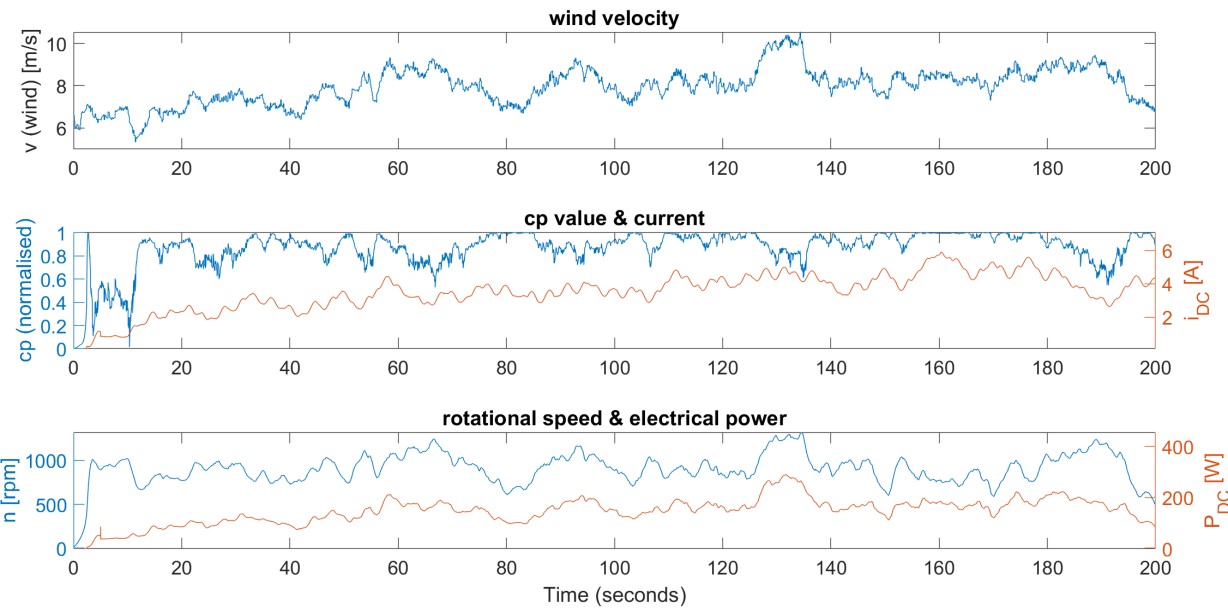

**Figure 7.** Simulation results with real wind data from collected in Kaiser-Wilhelm-Koog on the shore of the North Sea at a height of 10 m.

Following a successful cold-start, as in the nominal case, the controller reaches near optimal performance after 17 s. Throughout the demanding wind conditions, the controller

shows impressive robustness. The controller is capable of adapting to "slow" wind changes almost instantaneously. A cross validation of the controller with further wind data, not presented here, shows robustness and fast convergence almost everywhere. However, disruptive wind changes pose a challenge to control. The challenge was addressed by limiting the amplitude of high-passed power signals using saturation.

For strong and fast wind changes, as in the time frame $130\,s < t < 135\,s$, the ES controller deteriorates temporarily because the response time is too large ($\approx 14\,s$) to adapt in the meantime. In the time frames $60\,s < t < 70\,s$ and $180\,s < t < 190\,s$, we observe deteriorating controller performance. This is caused by a (casual) harmonic pattern in the wind velocity, with a frequency similar to the introduced perturbation signal. Thus, the gradient cannot be estimated reliably and the controller drifts. Note that, although the controller deteriorates temporarily, overall stability and a steady-state optimum are guaranteed.

## 5. Discussion

Tracking the maximum power point for wind turbines is in general a nontrivial challenge, requiring persistence of excitation, online estimation of the power map, and robust and stable control all together. As shown, extremum seeking is very well suited to addressing these challenges.

In this paper, an adaptive control to the recently developed wind turbine MOWEA was successfully applied. Key controller design criteria were fast adaption and robustness with respect to real wind conditions to harvest maximum power in urban areas.

To this end, a comprehensive aero-electromechanical model of the wind turbine under study including basic control was proposed. The derived model is nonlinear and features a power output map that exhibits a maximum power point. The maximum power point however is neither constant nor known due to changing wind conditions and production variability.

The proposed control provides fast and robust adaptation to partial and full load regions of small turbines without the need to introduce costly and error-prone wind speed sensors. Only electrical values already measured were taken into account. As shown, the proposed approach provides excellent performance for nominal and noisy wind changing regimes. Furthermore, the performance was validated considering real wind data, including fast and disruptive turbulence, and gusts, and demonstrated cold-start capability for the overall control scheme. Being advantageous, extremum seeking is easy to implement with only a few key design parameters. The proposed control scheme is computationally efficient and can be easily employed onto a local micro-controller.

For demanding disturbance regimes, as observed in urban areas and as considered here, fast adaptation is key. Thus, the high-perturbation signal frequency chosen as well as increasing the gain in the integral controller both increase the speed of adaptation. While increasing gain reduces the robustness margin, choosing a higher perturbation frequency allows us to harvest faster wind gusts. Therefore, the maximum perturbation frequency was chosen (to not excite the internal dynamics), and then, the appropriate gain was tuned to obtain robust adaption for all the considered urban wind scenarios.

Future work will address adaptive control of a variable array of turbines stacked together.

**Author Contributions:** All authors equally contributed to this work. All authors have read and agreed to the published version of the manuscript.

**Funding:** This research received no external funding.

**Institutional Review Board Statement:** Not applicable.

**Informed Consent Statement:** Not applicable.

**Conflicts of Interest:** The authors declare no conflict of interest.

## Abbreviations

The following abbreviations are used in this manuscript:

| | |
|---|---|
| AC | Alternating Current |
| BLDC | Brushless Direct Current |
| DC | Direct Current |
| ES | Extremum Seeking |
| ESC | Extremum Seeking Control |
| LIDAR | Light Detection and Ranging |
| MOWEA | Modulare Windenergieanlagen GmbH |
| SWT | Small Wind Turbine |
| PI | Proportional Integral |

**Nomenclature**

| | |
|---|---|
| $C_{cr}$ | inner current controller |
| $C_{ES}$ | Extremum Seeking (ES) controller |
| $i_{DC}$ | DC current of the load circuit |
| $k_M$ | electric machine (generator) parameter |
| $P_{DC}$ | DC power at the load |
| $P_{AC}$ | AC power of the electric machine (generator)/AC–DC converter |
| $P_{r,R}$ | rated wind turbine power |
| $P_w$ | power of free air flow along a stream tube |
| $R$ | rotor radius |
| $T_g$ | generator torque |
| $T_r$ | rotor torque |
| $v$ | wind speed |
| $v_d$ | design wind speed also known as rated wind speed |
| $\lambda$ | tip speed ratio |
| $\rho$ | air density |
| $\tau_i$ | time constant of the inner current control loop |
| $\psi$ | electrical machine (generator) flux |
| $\omega_p$ | frequency of perturbation signal of ES controller |
| $\omega_H$ | cutoff frequency of ES controller |
| $\omega_r$ | rotor angular velocity (rotor speed) |
| $\omega_{r,R}$ | rated rotor angular velocity (rated rotor speed) |

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
