# Peer review of "Adaptive Extremum Seeking Control of Urban Area Wind Turbines"

_energies, doi:10.3390/en14051356_

Round 1
Reviewer 1 Report
It is an interesting paper providing theory and simulation using real-data set.
However, it needs more improvements before being published as a journal paper.
- The authors do not use the Third Person.
- Literature review is weak with very few references and nearly none recent.
- Figure 4: It would be nice to define the slope of slow changing and fast changing wind velocities.
- Figure 4: It would be nice to commend on the response time in regards to slow and fast changing wind velocities.
- Figure 6: in the range 60<t<70s, 130<t<135s and 180<t<190s the windspeed is fairly constant and the performance of the controller deteriorates. Please commend.
- It would be nice to see the comparison with another set(s) of parameters and fitted values as discussed on section 3.2 Control Parameters
Thank you.
Author Response
Dear Reviewer,
please find the answers to your comments in the attached pdf file.

Reviewer 2 Report
General comments
The authors propose, in the submitted paper energies-1096714, a paper about “Adaptive Extremum Seeking Control of Urban AreaWind Turbines”.
The proposed paper deals with an interesting research field: wind turbines in Urban Areas. In this paper the authors applied an Extremum seeking control scheme to the recently developed wind turbine MOWEA. To this end, they proposed a comprehensive aero-electromechanical model of the wind turbine under study including basic control. Next, the Extremum seeking control scheme was adapted to the system. Several aspects to increase adaptation speed are highlighted, including a novel phase compensation. Finally, we validated the proposed approach considering real wind data, and thus demonstrated its fast and robust adaption capacity. The proposed control scheme is computationally efficient and can be easily implemented on the onboard wind turbine microcontroller.
The paper can be of great interest to energies readers, I would suggest some improvements to improve its global quality:
- Include a nomenclature section, if possible
- Avoid using “we”, the paper should use only scientific language
- The paper is too short, does not accomplishes with the requirements of a scientific paper and is rather a technical report. I would suggest extend it, if possible, and include, if possible, further literature review
- Remark and analyze the novelty of the paper, that is a key point, from my point of view
- Figures 4 and 5 should be analyzed before being presented, and include further analysis
- If possible, include a brief sensibility analysis
- I insist: remark the novelty of the research
- Provide a flow chart showing the research assumptions
The reviewer.
Author Response
Dear Reviewer,
Enclosed are the answers to your comments in the attached pdf file.

Reviewer 3 Report
An important (major) observation is related to the lack of comparisons with existing solutions of the same type (Extremum Seeking Control Wind Turbines). For example, a simple google search took me to https://www.researchgate.net/publication/333239030_Extremum_Seeking_Control_for_optimization_of_a_feed-forward_Pelton_turbine_speed_controller_in_a_fixed-displacement_hydraulic_wind_turbine_concept which seem to be a paper with similar theme or to https://iucrc.org/node/extremum-seeking-control-maximizing-wind-turbine-power-output which seems to be a commercial solution that has something similar with your solution presented in the paper. Add in the Introduction some references to existing works (or solutions that are already used) about extremum seeking control for wind turbines and mention the differences and novelty elements brought by your proposal compared to works of the type mentioned above. In this way the reader will have a clear picture of the strengths of the paper compared to other works or solutions.
In what environment was the system designed and simulated ? Describe the tools that were used to build the system and get the results for this paper. I consider it an important aspect to specify that will give readers an important reference about the results.
Author Response

(The authors gave the same response as above.)

Round 2
Reviewer 1 Report
Thank you for addressing my suggestions and concerns. I like the fact that the authors have included real data for testing the algorithm.
Accept in present form.
Thank you
Reviewer 2 Report
The authors have addressed the requirements and now the paper is ready for publication.
Sincerely, the reviewer.
Reviewer 3 Report
The authors responded to my comments. I have no comment. From my point of view, paper can be published in present form.